# Diversity of Summer Weed Communities in Response to Different Plum Orchard Floor Management in-Row

## Jerzy Lisek

The National Institute of Horticultural Research, Konstytucji 3 Maja 1/3 Str., 96-100 Skierniewice, Poland; jerzy.lisek@inhort.pl

**Abstract:** The effect of five methods of in-row weed management on the species composition and diversity of summer weed communities in a plum orchard was evaluated. Different methods of orchard floor management (OFM) were implemented for seven consecutive years from 2009 to 2015. *Festuca rubra* L. ssp. *rubra*–rhizomatous perennial grass was sown as a cover crop in the alleys of the orchard, in the tree planting year. In the seventh year of OFM implementation, the treatments were ranked according to the decreasing value of the Shannon–Wiener floristic diversity index as follows: tillage, post-emergence herbicides spraying, mowing, mulch, and weedy control. The highest value of Simpson dominance index was found in the control treatment. In plots with such treatments as control, mowing, tillage, and mulch, the dominant species was *F. rubra*. This meant that the rhizomatous cover crop from the alleys penetrated and affected the in-row flora. Vegetation of mulched plots was characterized by low value of density and soil cover. The obtained results indicated that the flora developing in the control, sprayed with post-emergence herbicides, tilled and mowed plots had greater potential to provide ecosystem services, than the flora of mulched plots.

**Keywords:** spontaneous flora; Shannon–Wiener diversity index; Simpson dominance index; *Prunus domestica* L.; *Festuca rubra* L. ssp. *rubra*





## 1. Introduction

The area of orchards in the European Union, including pome and stone fruits, citrus and olives, is approximately 6 million hectares [1]. These cultivated plants are of great economic, environmental, and landscape importance. Fruit trees require effective weed management, which is carried out by various methods, such as herbicide treatment, tillage, mowing [2], flame weeding, mulching [3], and cover crops, including living mulch [4–6]. Weeds pose threats to crops, such as competition for light, water, nutrients, allelopathic effect and increase in risk caused by diseases and pests, including rodents and spring frosts during the flowering of fruit trees [7,8]. Spontaneous vegetation also brings benefits, referred to as ecosystem services, such as increasing biodiversity, creating a habitat for beneficial organisms, providing food for bees and other pollinators, protecting the soil from erosion, salinity and mechanical compaction, reducing soil nutrient leaching, increasing the organic matter content in the soil, landscape, and ornamental functions [8–11]. Considering the environmental role of weeds, the European Union pursues a policy aimed at increasing the number of wild plants in the agricultural landscape [12]. Botanical diversity supports the efficient performance of ecosystem services by weed communities and is monitored in different time horizons, in arable [13], horticultural [14], and perennial industrial [15] crops. The comparison shows that in the conditions of the Indian Western Himalayas, weed species biodiversity in horticulture (apple and vegetable gardens) is greater than in arable crops [16]. A similar relationship between permanent crops (orchards and vineyards) and arable plants (cereals and row crops) was found on the Istrian peninsula in Croatia [14]. The diversity of weed species depends on environmental conditions, landscape structure and agricultural practices [17–19]. In arable crops, the main factor limiting the diversity of

weeds and selecting a specific set of species is the intensification of agricultural measures, primarily tillage, the use of herbicides and mineral fertilizers [20–23]. The intensification of agricultural practices may lead to a reduction in the diversity of orchard and vineyard flora in the long term [14]. The effect of various orchard floor management (OFM) methods on weed diversity was evaluated in orchards with different tree species, such as apple [24], apricot [25], fig [26], almond [27], and olives [28,29]. Closely related to the subject of the present studies are also the results concerning the effects of weed management on the flora diversity obtained in vineyards [30–33], where, similarly to orchards, an alley cropping system is maintained.

The hypothesis of the research was that the method of OFM leads to changes in agro-phytocenoses, important both for the environment, due to the ecosystem services provided by the flora, and for the fruit grower, who must protect trees from weed disservices, taking into account practical possibilities and economic realities. The aim of the study was to evaluate the multi-year implementation of selected OFM methods on the diversity of weed communities in a plum orchard, including the dynamics of changes in successive years, to indicate the methods that best preserve the initial diversity and the weak points of the selected methods.

## 2. Materials and Methods

### 2.1. Experimental Site, Material and Design

The field experiment was conducted in the Experimental Orchard of the National Research Institute of Horticulture in Dąbrowice, Central Poland (51°55′ N, 20°06′ E, 145 m a.s.l.). Orchard soil type, with neutral reaction (pH 6.5 in potassium chloride) and 2.3% of organic matter, was classified as luvisol, according to the international soil classification system [34]. Prior to the establishment of the orchard, forecrops–cereals and mustard—were cultivated for three years, but earlier, the area had been used for intensive orchards for over 30 years. The experimental plots were the internal part of a large, uniform plum orchard unit with an area of 2 ha (about 2500 trees) to limit the edge effect. Plum trees grew inside the Experimental Orchard with an area of 70 ha. Around it were commercial orchards, farmland with arable crops and ruderal places—villages and roads. One-year-old plum trees of the cultivar 'Valjevka' (*Prunus domestica* L.), grafted on Myrobalan seedlings (*Prunus cerasifera* Ehrh. *var*. *divaricata* Ledeb.), were planted in the spring of 2008. The trees were spaced at 2 m in rows and 4 m between rows. Trees trained to the central leader spindle system were drip irrigated. Soil water potential was kept between 0 and −0.02 MPa at a depth of 0.2 m, according to the reading on the tensiometers. Mineral fertilization—nitrogen (N): 50 kg/ha in 2008 and 2009, 30 kg/ha in 2010–2014, 15 kg/ha in 2015 (as ammonium nitrate); phosphorus (P): 50 kg/ha in 2008 (triple superphosphate); potassium (K): 75 kg/ha in 2008, 50 kg/ha in 2010, 50 kg/ha in 2012 (potassium chloride) and plant protection (three fungicide treatments against brown rot disease and three insecticide treatments against plum sawfly and plum moth per year) were carried out according to current recommendation for commercial orchards. Perennial, rhizomatous grass *Festuca rubra* L. ssp. *rubra* (red fescue), cultivar 'Leo', hereinafter referred to as *F. rubra*, was sowed in the inter-rows in September 2008. For seven consecutive years, from 2009 to 2015, the following methods of orchard floor management (OFM) in-row trees were introduced:

1. Control with limited weeding (manual weeding in spring within a radius of 0.5 from the tree trunk);
2. Spraying with post-emergence herbicides (glyphosate—two treatments per year at the rate of 2.88 kg a.i./ha in May and in the second half of August; glufosinate ammonium—one treatment, 0.6 kg a.i./ha in mid-June);
3. Mulching with organic waste—cereal straw with 2-yers-old compost from plant wastes in a volume ratio of 2:1 (layer of about 10 cm, filled in every 2 years, which was enough to effectively reduce the emergence of weeds);

4.　Tillage—mechanical soil cultivation with the use of rotary cultivators and hoe—three times from the beginning of May to August, on average every six weeks;

5.　Weed mowing—3 times between May and September. Mowing reduces weed growth less than tillage and herbicides. The last mowing was carried out about two weeks after tillage and herbicide spraying to limit weed regrowth closer to the onset of winter. Strong weed infestation in autumn attracts rodents.

Treatments were applied in completely randomized blocks with 4 replications (blocks) and 5 trees on the plots (20 trees per treatment). The width of the plots was 2 m, and their area was 20 m$^2$.

The study was carried out in the conditions of the temperate climate, intermediate between maritime and continental. During the study period, the average air temperature was 8.6 °C. January was the coldest month (−2.5 °C), and July was the hottest (19.4 °C). Average annual precipitation was 496 mm and ranged between 316 mm (in 2015) and 680 mm (in 2010).

*2.2. Measurements and Analyses*

2.2.1. Phytosociological Relevés

Relevé in phytosociology is a sample site in which all the plant species are described and documented. Data on plant occurrence of particular weed species, their share and importance in soil cover, were collected between 20 July and 10 August each year from 2008 to 2015, according to the Braun-Blanquet method [35], modified to the plot experiment design. Both early season and late-season weeds were present at the time of the survey. The ecosystem services provided by the summer vegetation are particularly important for biodiversity. Although the flora in the experiment was assessed in three terms, one mid-summer survey was chosen for the presentation, which enabled a precise and clear interpretation of the results. Weeds were additionally divided into groups–monocotyledonous (grassy) or dicotyledonous (broad-leaved); short-lived (annual + biennial) and perennial. The relevés with an area of 20 m$^2$ coincided with the experimental plots, therefore they are referred to as plots and were located under the tree canopies. In the year of establishing the orchard, hereinafter referred to as the base year, 20 plots were recorded, 5 in each block. In the first year of OFM implementation, the full cycle of interventions has not yet been completed. Weed response to OFM was studied starting from the second year of treatment differentiation, i.e., from the third year after planting the trees. In each plot, four sample squares, each with an area of 1m$^2$ (16 squares per treatment) were randomly placed, at least 0.5 m from tree trunks. Plots data were calculated as the mean of 4 sample squares.

2.2.2. Phytosociological Stability (S)

S was expressed on a 5-point scale. The stability classes according to the percentage of sample squares in which a given species was found were as follows: 5—81–100%, 4—61–80%, 3—41–60%, 2—21–40%, 1—1–20%.

2.2.3. The Cover Factor (CF)

CF was determined according to the formula:

$$CF = \frac{\sum CP_i}{N} \times 100$$

where: CP$_i$—percentage cover by i-th plant species in the sample square in which i-th species occurs; N—total number of sample squares.

By adding the cover factors for the various short-lived species and the perennials, the percentage share of these two groups in covering the soil with weeds was determined.

2.2.4. Weed Infestation Rate

The following importance classes were distinguished [36]:
I—very high: S = 5 or 4, CF > 1000;

II—high: S = 5 or 4, CF = 501–1000 or S = 3, CF > 750;
III—moderate: S = 5 or 4, CF = 251–500 or S = 3, CF = 501–750;
IV—low: S = 5 or 4, CF = 51–250 or S = 3, CF = 251–500;
V—sporadic: other lower stability classes and cover factors.

2.2.5. Diversity of Weed Communities

Diversity was compared for the OFM methods with the following indices:

- Shannon–Wiener diversity index–*H*′ [37]

$$H' = -\sum_{i=1}^{s} p_i \ln p_i,$$

- Simpson dominance index–*D* [38]

$$D = \sum_{i=1}^{s} (p_i)^2,$$

where: $p_i = n_i/N$, $n_i$—number of individuals of the i-th species; N—total number of individuals; s—number of species.

The Shannon–Wiener diversity index (*H*′) increases with the number of weed species in the community and the degree of equalization in their numbers. *H*′ values range from 0 to 5, usually ranging from 1.5 to 3.5, rarely above 4.5. The Simpson dominance index (*D*) takes on values in the range (0; 1>, and when the value reaches 1, it means there is no diversity (a single-species community). The data on the number of each species, necessary for the calculation of the indices, were collected from sample squares with an area of 1.0 m$^2$, as shown in Section 2.2.1. Weeds were identified by the author. Latin plant names follow the Polish national botanical Key book [39]

*2.3. Statistical Analysis*

Results concerning number of weed species in one plot, total number of weeds per m$^2$, the Shannon–Wiener diversity index (*H*′) value, and the Simpson dominance index (*D*) value, were analyzed statistically using analysis of variance. The significance of the means was evaluated using Duncan's test at 5% level. Statistical analyses were performed using the package STATISTICA 10.0 (StatSoft Inc., Tulsa, OK, USA).

**3. Results**

To show the effect of multi-year OFM on the flora, data on species composition, density and weed cover were compiled for the base year and the seventh year of OFM implementation.

*3.1. Weed Species Number, Density and Cover*

The comparison of selected weed infestation parameters showed that in the base year, the total number of weed species (36), the average number of species in one plot (21.5), and weed density (188.5) were higher than in the plots of each treatment, determined in the seventh year of OFM implementation, i.e., eight years later (Table 1). The lowest values of the three mentioned parameters were characteristic of the mulched plots, where the total number of species reached 14, the mean number of species per plot was 7.75, and the mean density of weeds was 18.4 pcs/m$^2$. In the base year, weed cover was dominated by short-lived species, which accounted for 94.9% of the cover (Table 1). In the seventh year of OFM implementation, the share of short-lived weeds in weed infestation was highest in the herbicide plots (88%) and lowest in weed control plots (3.6%). In the last year of the study, the relative share of grasses in the weed cover was higher in all treatments than in the base year, and the largest share (84.8%) was in the control plots (Table 1).

**Table 1.** In-row weed flora in the base year (2008) and in the seventh year of OFM implementation (2015).

| Feature | Base Year | Seventh Year of Implementation | | | | |
|---|---|---|---|---|---|---|
| | | Control | Herbicides | Mulch | Tillage | Mowing |
| Total number of weed species | 36 | 20 | 21 | 14 | 22 | 19 |
| Number of short-lived species | 25 | 11 | 14 | 4 | 14 | 12 |
| Number of perennial species | 11 | 9 | 7 | 10 | 8 | 7 |
| Number of broad-leaved species | 32 | 16 | 18 | 12 | 17 | 17 |
| Number of grassy species | 4 | 4 | 3 | 3 | 5 | 2 |
| Total CF | 9692 | 9971 | 8433 | 1188 | 8890 | 9183 |
| Total CF of short-lived weeds | 9196 | 358 | 7425 | 272 | 3386 | 2040 |
| Total CF of perennial weeds | 496 | 9613 | 1008 | 916 | 5504 | 7143 |
| Share of short-lived species in weed cover (%) | 94.9 | 3.6 | 88.0 | 22.9 | 38.1 | 22.2 |
| Share of perennial species in weed cover (%) | 5.1 | 96.4 | 12.0 | 77.1 | 61.9 | 77.8 |
| Share of broad-leaved species in weed cover (%) | 85.8 | 14.6 | 72.0 | 26.6 | 58.1 | 30.5 |
| Share of grassy species in weed cover (%) | 13.1 | 84.8 | 26.5 | 60.9 | 41.9 | 69.5 |
| Mean number of weed species in one plot | 21.5 ± 2.52 [d] | 11.25 ± 1.26 [b] | 14.5 ± 1.91 [c] | 7.75 ± 0.5 [a] | 14.0 ± 0.82 [c] | 12.25 ± 1.26 [b] |
| Weed density (pcs m$^2$) | 188.5 ± 9.75 [d] | 138.8 ± 6.67 [c] | 121.9 ± 5.63 [b] | 18.4 ± 2.01 [a] | 121.6±7.69 [b] | 143.7 ± 5.10 [c] |

Means followed by the same letter do not differ significantly at $p$ = 0.05. Values with the prefix ± represent standard deviation.

### 3.2. Weed Infestation Rate

In the base year, four dominant species of short-lived weeds were distinguished–*Chenopodium album, Capsella bursa-pastoris, Stellaria media,* and *Poa annua,* which were characterized by a very high infestation rate (class I) and one species–*Polygonum aviculare* occurring in II class with a high infestation rate (Table 2). After seven years of varied OFM, significant differences were found among the dominant species in the plots of individual treatments. The control plots were clearly dominated by two perennial species with a very high infestation rate, *Festuca rubra* ssp. *rubra* and *Epilobium ciliatum* (Table 3). Two short-lived species–*Stellaria media* and *Poa annua*—belonged to the I class of weed infestation in herbicide plots (Table 4). In the mulched plots, none of the species occurred at a very high infestation rate, and the dominant species turned out to be *F. rubra,* occurring in a II class infestation (Table 5). Three perennial species–*F. rubra, Rumex acetosella,* and *Elymus repens*—in class I of infestation, and weeds in a class II infestation included perennial–*Taraxacum officinale*—and short-lived–*Chenopodium album, Poa annua,* and *Stellaria media*—dominated the tilled plots (Table 6). *F. rubra* clearly dominated the mowed plots and 3 species–*Taraxacum officinale, Bromus hordeaceus,* and *Crepis biennis*–belonged to a class II infestation (Table 7). After 7 years of OFM implementation, the dominant species in the plots of all treatments, except for herbicides, was the *F. rubra*. Its occurrence was found under the tree canopy for the first time in the second year of OFM implementation, and from the fourth year it was more and more numerous, especially in control plots.

### 3.3. Shannon–Wiener Diversity Index (H′)

After seven years of OFM implementation, only one treatment with soil tillage did not significantly differ in the value of the in-row flora *H′* index, compared to the baseline value, which was 2.310 (Table 8). For all other treatments, the value of this index was significantly lower than the baseline value. The sequence of treatments according to the decreasing value of *H′* was as follows: tillage (2.285), herbicides (1.770), mowing (1.536), mulch (1.338), and weedy control (0.817). This means that the last of the treatments was characterized by the smallest biodiversity of flora. Differences between all OFM variants in the seventh year of implementation were statistically significant. In individual years of the assessment, both the value of *H′* and the relationship between treatments regarding the value of *H′* changed. In the second year of OFM implementation, the *H′* value for all treatments, except for control, was significantly lower than the baseline value. In the third, fourth, and fifth year, the value of *H′* was significantly lower than the baseline value for all evaluated methods of OFM. In the sixth year of OFM implementation, the relationships between treatments were shaped in the same way as in the seventh year of implementing the five OFM methods.

**Table 2.** In-row weed flora in the base year (2008).

| Species | Phytosociological Stability (S) | Cover Factor (CF) | Weed Infestation Rate (Class) | Mean Number of Weeds (Pcs m$^2$) |
|---|---|---|---|---|
| **Short-lived** | | | | |
| *Chenopodium album* L. | 5 | 2146 | I | 44.3 |
| *Capsella bursa-pastoris* (L.) Medik. | 5 | 1772 | I | 34.8 |
| *Stellaria media* (L.) Vill. | 5 | 1554 | I | 30.5 |
| *Poa annua* L.(G) | 5 | 1116 | I | 26.3 |
| *Polygonum aviculare* L. | 5 | 574 | II | 10.7 |
| *Senecio vulgaris* L. | 5 | 376 | III | 7.0 |
| *Polygonum persicaria* L. (*P. maculosa* Gray) | 4 | 224 | IV | 6.3 |
| *Matricaria maritima* L. ssp. *inodora* (L.) Dostál | 4 | 212 | IV | 4.3 |
| *Galium aparine* L. | 4 | 158 | IV | 3.0 |
| *Fallopia convolvulus* (L.) A. Löve | 4 | 148 | IV | 2.8 |
| *Veronica persica* Poir. | 4 | 145 | IV | 2.3 |
| *Echinochloa crus-galli* (L.) P. Beauv. (G) | 3 | 128 | V | 2.0 |
| *Veronica arvensis* L. | 2 | 106 | V | 1.5 |
| *Viola arvensis* Murr. | 2 | 101 | V | 1.3 |
| *Lamium purpureum* L. | 2 | 98 | V | 1.0 |
| *Geranium pusillum* Burm. F. ex L. | 2 | 82 | V | 1.0 |
| *Chamomilla suaveolens* (Pursh) Rydb. | 1 | 64 | V | 0.5 |
| *Amaranthus retroflexus* L. | 1 | 42 | V | 0.5 |
| *Conyza canadensis* (L.) Cronq. | 1 | 41 | V | 0.5 |
| *Crepis capillaris* (L.) Wallr. | 1 | 38 | V | 0.3 |
| *Crepis biennis* L. | 1 | 32 | V | 0.2 |
| *Vicia villosa* Roth. | 1 | 12 | V | <0.05 |
| *Bromus hordeaceus* L. (G) | 1 | 10 | V | <0.05 |
| *Erodium cicutarium* (L.) L'Her. | 1 | 9 | V | <0.05 |
| *Atriplex patula* L. | 1 | 8 | V | <0.05 |
| **Perennial** | | | | |
| *Equisetum arvense* L. | 3 | 114 | V | 2.3 |
| *Taraxacum officinale* F. H. Wigg. | 3 | 102 | V | 2.0 |
| *Cerastium holosteoides* Fr. em. Hyl. | 3 | 76 | V | 1.3 |
| *Trifolium repens* L. | 2 | 44 | V | 0.5 |
| *Plantago major* L. | 1 | 42 | V | 0.5 |
| *Epilobium ciliatum* Raf. | 1 | 38 | V | 0.5 |
| *Convolvulus arvensis* L. | 1 | 26 | V | 0.2 |
| *Cirsium arvense* (L.) Scop | 1 | 18 | V | 0.1 |
| *Rumex acetosella* L. | 1 | 16 | V | <0.05 |
| *Elymus repens* (L.) Gould (G) | 1 | 12 | V | <0.05 |
| *Urtica dioica* L. | 1 | 8 | V | <0.05 |

G—grassy species.

### 3.4. Simpson Dominance Index (D)

The baseline value of index *D* was 0.139. In the seventh year of OFM implementation, the lowest value of the indice, not significantly different from the baseline value, was obtained for treatment tillage (Table 9). The value of *D* for all other treatments was significantly higher than the baseline value. Treatments were ranked according to increasing *D* index values in the following order: tillage (0.123), herbicides (0.269), mulch (0.365), mowing (0.381), and control (0.681). Differences between mulch and mowing treatments were insignificant. Weedy control was the treatment with the most marked species dominance among synanthropic plants of tree understory, in the last year of the assessment. In the fifth, sixth, and seventh year of OFM implementation, the relationships in the significance of differences between treatments were the same.

**Table 3.** In-row weed flora in the seventh year of OFM implementation (2015)–control.

| Species | Phytosociological Stability (S) | Cover Factor (CF) | Weed Infestation Rate (Class) | Mean Number of Weeds (Pcs m²) |
|---|---|---|---|---|
| Short-lived | | | | |
| *Matricaria maritima* L. ssp. *inodora* (L.) Dostál | 4 | 85 | IV | 1.8 |
| *Galium aparine* L. | 4 | 62 | IV | 1.3 |
| *Crepis biennis* L. | 4 | 54 | IV | 1.0 |
| *Conyza canadensis* (L.) Cronq. | 4 | 48 | V | 0.8 |
| *Bromus hordeaceus* L.(G) | 3 | 30 | V | 0.5 |
| *Fallopia convolvulus* (L.) A. Löve | 3 | 22 | V | 0.3 |
| *Stellaria media* (L.) Vill. | 2 | 19 | V | 0.2 |
| *Chenopodium album* L. | 2 | 12 | V | 0.1 |
| *Echinochloa crus-galli* (L.) P. Beauv. (G) | 1 | 10 | V | 0.1 |
| *Geranium pusillum* Burm. F. ex L. | 1 | 9 | V | 0.1 |
| *Tragopogon pratensis* L. | 1 | 7 | V | < 0.05 |
| Perennial | | | | |
| *Festuca rubra* L. ssp. *rubra* (G) | 5 | 8452 | I | 115.8 |
| *Epilobium ciliatum* Raf. | 5 | 508 | I | 6.5 |
| *Rumex acetosella* L. | 5 | 316 | III | 4.3 |
| *Taraxacum officinale* F. H. Wigg. | 5 | 89 | IV | 1.8 |
| *Cerastium holosteoides* Fr. em. Hyl. | 5 | 82 | IV | 1.5 |
| *Equisetum arvense* L. | 4 | 64 | IV | 1.3 |
| *Convolvulus arvensis* L. | 3 | 55 | V | 0.8 |
| *Elymus repens* (L.) Gould (G) | 2 | 28 | V | 0.5 |
| *Artemisia vulgaris* L. | 1 | 19 | V | <0.1 |

G—grassy species.

**Table 4.** In-row weed flora in the seventh year of OFM implementation (2015)—herbicides.

| Species | Phytosociological Stability (S) | Cover Factor (CF) | Weed Infestation Rate (Class) | Mean Number of Weeds (Pcs·m²) |
|---|---|---|---|---|
| Short-lived | | | | |
| *Stellaria media* (L.) Vill. | 5 | 4080 | I | 52.3 |
| *Poa annua* L. (G) | 5 | 1950 | I | 27.5 |
| *Lamium purpureum* L. | 5 | 375 | III | 6.8 |
| *Conyza canadensis* (L.) Cronq. | 5 | 280 | III | 4.8 |
| *Echinochloa crus-galli* (L.) P. Beauv. (G) | 5 | 212 | IV | 4.5 |
| *Chenopodium album* L. | 4 | 169 | IV | 3.3 |
| *Capsella bursa-pastoris* (L.) Medik. | 4 | 115 | IV | 2.5 |
| *Bromus hordeaceus* L. (G) | 3 | 76 | V | 1.5 |
| *Viola arvensis* Murr. | 3 | 59 | V | 1.0 |
| *Veronica arvensis* L. | 4 | 38 | V | 0.8 |
| *Polygonum aviculare* L. | 2 | 29 | V | 0.5 |
| *Fallopia convolvulus* (L.) A. Löve | 1 | 19 | V | 0.3 |
| *Geranium pusillum* Burm. F. ex L. | 1 | 15 | V | 0.3 |
| *Galium aparine* L. | 1 | 8 | V | 0.1 |
| Perennial | | | | |
| *Taraxacum officinale* F. H. Wigg. | 5 | 460 | III | 7.3 |
| *Epilobium ciliatum* Raf. | 5 | 302 | III | 5.3 |
| *Equisetum arvense* L. | 5 | 124 | IV | 2.5 |
| *Cerastium holosteoides* Fr. em. Hyl. | 3 | 62 | V | 1.3 |
| *Trifolium repens* L. | 1 | 26 | V | 0.5 |
| *Rumex acetosella* L. | 1 | 18 | V | <0.1 |
| *Urtica dioica* L. | 1 | 16 | V | <0.1 |

G—grassy species.

**Table 5.** In-row weed flora in the seventh year of OFM implementation (2015)–mulch.

| Species | Phytosociological Stability (S) | Cover Factor (CF) | Weed Infestation Rate (Class) | Mean Number of Weeds (Pcs·m$^2$) |
|---|---|---|---|---|
| Short-lived | | | | |
| *Galium aparine* L. | 5 | 152 | IV | 2.3 |
| *Atriplex patula* L. | 4 | 69 | IV | 1.0 |
| *Chenopodium album* L. | 4 | 42 | V | 0.5 |
| *Fallopia convolvulus* (L.) A. Löve | 3 | 9 | V | 0.1 |
| Perennial | | | | |
| *Festuca rubra* L. ssp. rubra (G) | 5 | 640 | II | 10.5 |
| *Urtica dioica* L. | 4 | 115 | IV | 1.5 |
| *Elymus repens* (L.) Gould (G) | 4 | 84 | IV | 1.3 |
| *Convolvulus arvensis* L. | 4 | 18 | V | 0.3 |
| *Artemisia vulgaris* L. | 3 | 18 | V | 0.3 |
| *Epilobium ciliatum* Raf. | 2 | 15 | V | 0.2 |
| *Malva neglecta* L. | 2 | 8 | V | 0.1 |
| *Rumex crispus* L. | 1 | 8 | V | 0.1 |
| *Tanacetum vulgare* L. | 1 | 5 | V | <0.1 |
| *Calamagrostis epigejos* (L.) Roth (G) | 1 | 5 | V | <0.1 |

G—grassy species.

**Table 6.** In-row weed flora in the seventh year of OFM implementation (2015)–tillage.

| Species | Phytosociological Stability (S) | Cover Factor (CF) | Weed Infestation Rate (Class) | Mean Number of Weeds (Pcs·m$^2$) |
|---|---|---|---|---|
| Short-lived | | | | |
| *Chenopodium album* L. | 5 | 829 | II | 13.0 |
| *Poa annua* L. (G) | 5 | 628 | II | 11.5 |
| *Stellaria media* (L.) Vill. | 5 | 514 | II | 9.3 |
| *Conyza canadensis* (L.) Cronq. | 5 | 382 | III | 5.1 |
| *Bromus hordeaceus* L. (G) | 4 | 246 | IV | 3.0 |
| *Crepis biennis* L. | 4 | 198 | IV | 2.8 |
| *Echinochloa crus-galli* (L.) P. Beauv. (G) | 4 | 154 | IV | 2.5 |
| *Polygonum persicaria* L. (*P. maculosa* Gray) | 3 | 110 | V | 1.8 |
| *Polygonum aviculare* L. | 3 | 94 | V | 1.3 |
| *Galium aparine* L. | 2 | 81 | V | 0.8 |
| *Fallopia convolvulus* (L.) A. Löve | 2 | 63 | V | 0.5 |
| *Capsella bursa-pastoris* (L.) Medik. | 1 | 42 | V | 0.5 |
| *Senecio vulgaris* L | 1 | 25 | V | 0.3 |
| *Geranium pusillum* Burm. F. ex L. | 1 | 20 | V | 0.3 |
| Perennial | | | | |
| *Festuca rubra* L. ssp. *rubra* (G) | 5 | 1620 | I | 21.5 |
| *Rumex acetosella* L. | 5 | 1584 | I | 18.8 |
| *Elymus repens* (L.) Gould (G) | 5 | 1080 | I | 15.5 |
| *Taraxacum officinale* F. H. Wigg. | 5 | 572 | II | 5.3 |
| *Cerastium holosteoides* Fr. em. Hyl. | 4 | 346 | III | 4.5 |
| *Equisetum arvense* L. | 4 | 148 | IV | 1.8 |
| *Epilobium ciliatum* Raf. | 4 | 126 | IV | 1.5 |
| *Cirsium arvense* (L.) Scop. | 1 | 28 | V | <0.1 |

G—grassy species.

**Table 7.** In-row weed flora in the seventh year of OFM implementation (2015)–mowing.

| Species | Phytosociological Stability (S) | Cover Factor (CF) | Weed Infestation Rate (Class) | Mean Number of Weeds (Pcs·m²) |
|---|---|---|---|---|
| Short-lived | | | | |
| *Bromus hordeaceus* L. (G) | 5 | 742 | II | 11.8 |
| *Crepis biennis* L. | 5 | 516 | II | 9.0 |
| *Conyza canadensis* (L.) Cronq. | 5 | 294 | III | 4.3 |
| *Matricaria maritima* L. ssp. *inodora* (L.) Dostál | 4 | 156 | IV | 3.0 |
| *Stellaria media* (L.) Vill. | 3 | 82 | V | 1.8 |
| *Polygonum aviculare* L. | 2 | 76 | V | 1.5 |
| *Geranium pusillum* Burm. F. ex L. | 2 | 54 | V | 1.0 |
| *Lamium purpureum* L. | 1 | 42 | V | 0.8 |
| *Chenopodium album* L. | 1 | 40 | V | 0.8 |
| *Vicia villosa* Roth. | 1 | 22 | V | 0.2 |
| *Crepis capillaris* (L.) Wallr. | 1 | 10 | V | 0.1 |
| *Tragopogon pratensis* L. | 1 | 6 | V | <0.05 |
| Perennial | | | | |
| *Festuca rubra* L. ssp. *rubra* (G) | 5 | 5640 | I | 84.0 |
| *Taraxacum officinale* F. H. Wigg. | 5 | 512 | II | 9.5 |
| *Cerastium holosteoides* Fr. em. Hyl. | 5 | 454 | III | 7.3 |
| *Rumex acetosella* L. | 4 | 280 | III | 4.5 |
| *Epilobium ciliatum* Raf. | 4 | 196 | IV | 3.3 |
| *Cirsium arvense* (L.) Scop. | 1 | 52 | V | 0.8 |
| *Trifolium repens* L. | 1 | 9 | V | <0.1 |

G—grassy species.

**Table 8.** Shannon–Wiener in-row flora diversity index (*H′*) in response to OFM.

| Treatment | *H′* Value | | | | | |
|---|---|---|---|---|---|---|
| Base year | 2.310 ± 0.094 [c] | 2.310 ± 0.094 [c] | 2.310 ± 0.094 [d] | 2.310 ± 0.094 [d] | 2.310 ± 0.094 [e] | 2.310 ± 0.094 [e] |
| | **Year of OFM implementation** | | | | | |
| | **2nd** | **3rd** | **4th** | **5th** | **6th** | **7th** |
| Control | 2.165 ± 0.049 [c] | 1.747 ± 0.292 [b] | 1.602 ± 0.147 [b] | 1.277 ± 0.080 [a] | 1.055 ± 0.072 [a] | 0.817 ± 0.130 [a] |
| Herbicides | 1.897 ± 0.057 [b] | 1.820 ± 0.113 [b] | 1.786 ± 0.160 [b] | 1.781 ± 0.064 [b] | 1.777 ± 0.110 [d] | 1.770 ± 0.155 [d] |
| Mulch | 1.620 ± 0.259 [a] | 1.418 ± 0.293 [a] | 1.360 ± 0.054 [a] | 1.353 ± 0.049 [a] | 1.343 ± 0.057 [b] | 1.338 ± 0.119 [b] |
| Tillage | 1.873 ± 0.092 [b] | 1.928 ± 0.124 [b] | 2.059 ± 0.175 [c] | 2.140 ± 0.154 [c] | 2.183 ± 0.131 [e] | 2.285 ± 0.072 [e] |
| Mowing | 1.869 ± 0.095 [b] | 1.854 ± 0.059 [b] | 1.709 ± 0.068 [b] | 1.660 ± 0.045 [b] | 1.616 ± 0.051 [c] | 1.536 ± 0.077 [c] |

Means within column followed by the same letter do not differ significantly at *p* = 0.05. Values with the prefix ± represent standard deviation.

**Table 9.** Simpson in-row flora dominance index (*D*) in response to OFM.

| Treatment | *D* Value | | | | | |
|---|---|---|---|---|---|---|
| Base | 0.139 ± 0.019 a | 0.139 ± 0.019 a | 0.139 ± 0.019 a | 0.139 ± 0.019 a | 0.139 ± 0.019 a | 0.139 ± 0.019 a |
| | **Year of OFM implementation** | | | | | |
| | **2nd** | **3rd** | **4th** | **5th** | **6th** | **7th** |
| Control | 0.173 ± 0.009 [ab] | 0.226 ± 0.057 [b] | 0.316 ± 0.075 [bc] | 0.447 ± 0.045 [d] | 0.569 ± 0.026 [d] | 0.681 ± 0.064 [d] |
| Herbicides | 0.213 ± 0.016 [bc] | 0.203 ± 0.030 [ab] | 0.250 ± 0.074 [b] | 0.234 ± 0.033 [b] | 0.240 ± 0.041 [b] | 0.269 ± 0.026 [b] |
| Mulch | 0.241 ± 0.051 [c] | 0.372 ± 0.091 [c] | 0.381 ± 0.030 [c] | 0.329 ± 0.019 [c] | 0.330 ± 0.020 [c] | 0.365 ± 0.047 [c] |
| Tillage | 0.208 ± 0.030 [bc] | 0.215 ± 0.32 [ab] | 0.171 ± 0.035 [a] | 0.155 ± 0.030 [a] | 0.145 ± 0.027 [a] | 0.123 ± 0.010 [a] |
| Mowing | 0.245 ± 0.022 [c] | 0.195 ± 0.012 [ab] | 0.295 ± 0.022 [b] | 0.314 ± 0.035 [c] | 0.329 ± 0.021 [c] | 0.381 ± 0.022 [c] |

Means within column followed by the same letter do not differ significantly at *p* = 0.05. Values with the prefix ± represent standard deviation.

## 4. Discussion

The results of the present study indicate that the diversity of weed communities was strongly differentiated depending on the OFM method used, which is consistent with previous reports on this topic in apple, apricot, fig, almond, and olives orchards [24–29] and vineyards [30–33]. As in the apple orchard, this should be justified by the diverse spectrum of weeds controlled by each of the OFM methods and the competitive ability of the dominant flora species [6]. In the last year of the study, i.e., in the seventh year of OFM implementation, regardless of the treatment, the density of weeds and the number of species in one plot were significantly lower than in the base year. The post-emergence herbicides application did not result in a drastic reduction in the number of weeds, and in the long term, even favored a greater diversity of flora compared to the control. The lack of pre-emergence activity of the herbicides allowed for a quick recovery of secondary and primary (in the spring of the next season) weed infestation. The large number of weeds in the herbicide plots also resulted from the rich seed bank in the soil. This bank was supplemented by weeds developed between herbicide treatments and by the wind, and they were mainly seeds of weeds from the Asteracea (*Taraxacum officinale*, *Conyza canadensis*) and Onagraceae (*Epilobium ciliatum*) families. Additionally, some perennial weeds regrow after spraying with herbicides. The greatest reduction in the weed number and cover was achieved after the use of mulch, which prevented the germination of weed seeds. The unmown vegetation in the control effectively stopped falling plum leaves. In combination with drying weed shoots, they formed a mulch that limited the germination and development of short-lived weeds. While weed infestation with short-lived species prevailed in the base year, perennial species prevailed in the plots of four treatments–control, mulch, tillage, and mowing–during the final assessment. Intensive weeding treatments act as a management filter, i.e., they effectively eliminate specific species or groups of weeds [10,30]. This creates conditions for the development of species that are difficult to eliminate and is particularly visible in the absence of weed control methods rotation over a long period of time [10,30]. The most numerous short-lived weeds in herbicide and tilled plots were *Poa annua* and *Stellaria media*–a species with a long period of emergence—and in tilled plots, *Chenopodium album*–a species characterized by a large number of seeds per plant and long persistence of seeds in the soil [40]. Creeping weeds dominated both in plots where the soil was not disturbed (control, mulch, and mowing) and in tilled plots, which indicated that cutting of rhizomes was conducive to their proliferation. In the seventh year of OFM implementation, the relative share of grasses in the weed cover on plots of all treatments increased compared to the baseline year. The present research confirmed the tendency to promote the development of grasses after spraying with post-emergence herbicides and mowing, noted in an almond orchard [27], but the grasses on the control plots developed even stronger. Grasses tolerated mowing much better than dicotyledonous plants, which results from their biology [27]. Glufosinate ammonium, one of the herbicides used in the present study (second spraying), was more effective at controlling dicotyledonous weeds than grasses. Mulch was not a sufficient barrier to the development of creeping weeds, among which *F. rubra* was dominant. The obtained results were related to the specific composition of the flora and the strong pressure of the cover plant from the orchard alleys. In order to interpret the present results, it is necessary to consider the data on the dominant flora species, especially their share in cover after seven years of OFM implementation. In the base year, *F. rubra* was sown in the orchard alleys as a cover plant to reduce weed infestation, soil erosion, and compaction, to ensure easy passage of machines and nutrient recirculation. *F. rubra* plants appeared in the experimental plots in the second year of OFM implementation, and from the fourth year, their dominance progressed rapidly. The highest dynamics of plant development of this species were found in the control plots. In mulched and tilled plots, *F. rubra* occurred mainly on their edges facing the alleys.

The final value of the flora Shannon–Wiener's diversity index (*H'*) for treatments with active measures, ranged in present studies from 1.338 to 2.285, and for the control it

was 0.817. Taking into account the results of other agro-phyteconoses obtained by many authors [13–16,24,25,31,33,41], it should be recognized that the evaluated OFM methods maintained the species diversity of the synanthropic flora at a satisfactory level. For comparison, the value of weed $H'$ in cereals in southern Poland was 0.97–1.08 [13], and in perennial industrial plants in eastern Poland it ranged from 0.7 to 2.3, depending on the crop and season [15]. The relatively low diversity of flora in cereals resulted from the uniformity of habitat conditions and agriculture landscape, and the long-term dominance of cereals in the assessment sites [13]. Data from industrial plants referred to a variety of crop species, and the experiment place was surrounded by a diverse landscape [15]. The $H'$ value for flora in apple orchards was 0.67–1.18 in South Korea [41], 3.325 in Indian Western Himalayas [16], and 2.264 for orchards and vineyards on the Istrian peninsula in Croatia [14]. According to the cited authors, the intensification of production in orchards, related to the increase in their area and the frequency of activities, resulted in a decrease in orchard vegetation diversity. Large differences in the value of $H'$ result from different environmental conditions, agricultural structure, intensity of agricultural practices [13], and the date (season) of assessment [41]. It should be noted that the experimental orchard from the present research was located in the middle of a large complex with fruit crops, cultivated for many years with high intensity. Common species prevailed, and no species of high environmental value were found–endangered, rare, or endemic taxa—under the conditions of the present research, as well as in a homogenous olive-dominated landscape in South-East of Italy [29]. Valuable species may be more easily found if research is conducted at various locations across the region, close to semi-natural habitats such as South African vineyards [33]. Valuable plants are those species that especially promote functional agrobiodiversity, e.g., *Achillea millefolium* L., *Centaurea jacea* L., *Leucanthemum vulgare* Lam., *Lotus corniculatus* L., and *Trifolium pratense* L. [42].

The second indice characterizing the diversity of weed communities was Simpson dominance index ($D$). The highest value of $D$, among the evaluated treatments was characterized by control (0.681), which proves the strong dominance of a few species. For comparison, in other studies conducted in Poland, the $D$ value was 0.17–0.22 in cereal crops [13] and 0.13–0.45 in perennial industrial crops [15]. In the present studies, the lowest value of $D$ was obtained for tillage, which at the same time was characterized by the highest value of $H'$. With a similar number of weed species, the treatment with herbicides was characterized by a significantly lower $H'$ value and a higher $D$ value than tillage. This shows that the dominance of the most abundant species was more pronounced after herbicide application than after tillage. Herbicides had a stronger filtering effect, i.e., limiting the weed species diversity more strongly than tillage. The results of the present experiment, probably due to the specific species composition, did not confirm that herbicides and tillage are a stronger management filter than mowing [22]. The use of herbicides in the present study reduced the $H'$ value compared to tillage, as in South African vineyards [33] and in an apricot orchard in Turkey [25], but did not reduce diversity compared to the weed control, which has been noted in an apricot orchard [25] and in vineyards located in southern France [30]. When discussing the reports of other authors, the duration of the study should be taken into account. In the case of apricot orchard and vineyards, these were results from 2 to 3 years [25,30]. The present results covered seven years of diversified OFM, so with each year the importance of succession decreased, and the importance of the weed management method increased. After the second year of OFM implementation in the plum orchard, herbicides significantly reduced the flora diversity compared to the control, as reported by other authors [25,30]. With each subsequent year of research, the diversity in herbicide plots, represented as $H'$, slightly decreased.

Herbicides were the only effective management filter that completely eliminated *F. rubra* from weed cover in the tree rows. In Polish orchards, bunch-type grasses are usually sown in the inter-rows of the orchard, and post-emergence herbicides are used under the tree canopies. With such a model, the problem of cover plants penetrating tree understory practically does not occur. Under the conditions of the present experiment, the

rhizomatous groundcover crop from the alleys significantly influenced the diversity of in-row flora. The development of *F. rubra* on experimental plots raises the question of whether this species can be used in orchards as a so-called service crop that is grown to reduce weed pressure and provide ecosystem services without generating serious disservices and impairing fruit production. An example of service plants in the tree understory are perennial groundcovers, also referred to as living mulches [6]. Service plants, in addition to intercrops and cover crops, may also include some weeds [10,30,32]. *F. rubra* turned out to be a very effective competitor to weeds in the present research, especially in the control plots. In contrast, *Festuca ovina*, used in the understory of apple trees as a live mulch, did not effectively limit the development of perennial weeds [6]. Cover crops, as an effective competitor, can reduce flora diversity [24,31]. Perennial *Duchesnea indica* (Andrews) Focke (*Potentilla indica* (Andrews) Th. Wolf) used as a cover crop in a Chinese apple orchard reduced the flora $H'$ value by 53.8%, from approximately 1.8 to 0.9.within three years [24]. In Californian vineyards, the $H'$ value for treatments of cover crops–oat and/or legumes—or cover crops + tillage was 1.2–1.4, while for the treatment of resident weeds + tillage, it was 1.8 [31]. For comparison, in the present study, this value for tillage was 2.285 in the seventh year of implementation. The effect of spreading red fescue from the alleys can be compared to the effects of invasive plants, e.g., *Xanthium strumarium* L., which reduced the diversity of weed communities [43].

Whether *F. rubra* is a strong competitor to trees remains an important question. A study conducted in parallel with the present one showed that plum trees grafted on 'Myrobalan' seedlings (*Prunus cerasifera* Ehrh. var. *divaricata* Ledeb.) growing on the control plots, dominated by *F. rubra*, were characterized by a similar cumulative yield and productivity index as on the plots of other treatments [44]. In-row flora, including *F. rubra*, did not worsen the nutritional status of plum trees in the parallel study either [45].

Fracchiolla et al. [27] found that vegetation mowing and post-emergence herbicides promoted in the almond orchard the growth of sufficiently diversified and balanced flora that could lead to potential ecological services. The results of the present research suggest that such services can also be carried out by flora in tilled and control plots. In plots sprayed with post-emergence herbicides, mowed and tilled, the spontaneous flora regrowth was quick after the intervention and was not disturbed in the period from mid-September to May of the following year. Numerous weeds, such as *Stellaria media*, *Taraxacum officinale*, *Galium aparine*, and *Cirsium arvense*, which are considered to be melliferous plants [15], were found in the plots of the mentioned treatments and in the control. Mulching with natural materials is a good way to reduce weed infestation in organic orchards, as it is not associated with soil disturbance and chemical residues [3]. In the present study, mulching generated sufficient weed diversity, but due to poor development and low soil cover, spontaneous flora could not provide ecosystem services as well as other treatments.

From a practical point of view, the present research confirmed, in accordance with previous reports, the insufficient effectiveness of controlling perennial creeping weeds (*Elymus repens*, *Rumex acetosella*) by soil tillage [46] and mulching [3].

The results of the present research indicate the potential possibility of using *F. rubra* ssp. *rubra* sown in orchard alleys as spontaneous, self-propagating living mulch that limits weed infestation. Such application should be supplemented with studies on the tolerance of fruit trees to the presence of *F. rubra* in the understory, monitoring the occurrence of rodents and combining with other OFM methods. Post-emergence herbicide application followed by flora mowing may be needed in young orchards to protect poorly growing trees from early and strong *F. rubra* competition.

## 5. Conclusions

The OFM in-row strongly influenced the floristic diversity of summer weed communities in the plum orchard. The values of the diversity indices and the relationships between the treatments regarding the values of these indices changed during the subsequent years of the assessment. After seven years of OFM implementation, the treatments were ranked

according to the decreasing value of the Shannon–Wiener diversity index as follows: tillage, post-emergence herbicides spraying, mowing, mulch, and control. The highest value of Simpson dominance index was found in the control treatment. *F. rubra* ssp. *rubra* was the dominant species in plots with treatments such as control, mowing, tillage, and mulch. This rhizomatous perennial grass, which was sown in the alleys of the orchard in the year of its establishment, penetrated and influenced in-row flora. Mulch efficiently reduced weed infestation, and therefore the vegetation of mulched plots was characterized by a low value of density and soil cover. The obtained results indicated that the flora overgrowing the control, sprayed with post-emergence herbicides, tilled, and mowed plots had greater potential to provide ecosystem services than the flora of mulched plots.

**Funding:** The publication was financed (co-financed) by the Ministry of Education and Science of Poland as part of the statutory activities of The National Institute of Horticultural Research, Project No. 3.2.1., "The response of synanthropic orchard flora and plum trees to chosen systems of weed control and soil management".

**Data Availability Statement:** Data are contained within the article.

**Conflicts of Interest:** The author declares no conflict of interest.

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
