# Peer review of "Diversity of Summer Weed Communities in Response to Different Plum Orchard Floor Management in-Row"

_agronomy, doi:10.3390/agronomy13051421_

Round 1

Reviewer 1 Report

The manuscript shows results of a multi-year study of weed diversity in a plum orchard. The methodology has to be explained better and the discussion need to improve, there are some points to development about the results obtained in this research. Authors mention data from other similar studies but they not compare or discuss with their results. 

The observations and comments are in the file attached.

Author Response

Skierniewice, 16.05.2023

Response to the Reviewer's 1 comments

The author wish to thank the Reviewer for your valuable comments that will improve the quality of the reviewed manuscript " Diversity of Summer Weed Communities in Response to Different Plum Orchard Floor Management in-row” (agronomy- 2391774). I have completed the manuscript  according to the Reviewers comments. The text has been corrected or supplemented to take into account the comments of the Reviewer.

First, I would like to inform you that during the processing of the text in the editor's office, an error in the numbering of references was made. In the original MS Word template as well as in the original PDF version sent to the editor's office, the number of references was 42. In items 7, 38, 42, links to publications were misread as references. The Spacebar or Enter key were unnecessarily used (probably my fault). The wrong numbering was the cause of the wrong citation.

It is advisable to specify the type of climate. Publications on orchards and vineyards routinely include information on temperatures in coldest and hottest months. In the Discussion I added an explanation of the reason for the flora diversity variability in weed infestation after seven years and the high CF value for herbicides. I agree with the reviewer that some of the sentences in the discussion are a repetition of the results. Some have been removed, and some have been left for the consistency of the text, because their removal will make discussion with other authors more difficult. The reviewer suggests very detailed comparisons of the values ​​of diversity indicators obtained in the present study and by other authors. Data from present study are included in the results, and sometimes it is difficult to compare them, because the treatments in the own study and in the experiments of other authors were not the same. The Discussion omitted some cited studies, focusing primarily on orchards and vineyards. From the research on arable crops, those that referred to Poland or contained the values ​​of similar diversity indices were used for discussion.

 Ad some detailed commentsAd comment l. 257. Changed as recommended by the ReviewerAd comment l. 296. I was able to compare results only with available articles, and these usually contain results relating to  a shorter period of time.Ad comment l. 326. Weeds as service plants. Weeds can be also service plants, as bring the same benefits, described in Introduction.

Ad comment l. 327. Festuca ovina ist mentioned for the first time, the whole name is better.

Yours Sincerely

Jerzy Lisek

Reviewer 2 Report

This paper compares five methods of weed management in a fruit tree orchard across eight years following initial cover of the alleys with Festuca rubra.  Mulching provided the least soil cover and was found unsatisfactory, while the other four methods (mowing, herbicide, control, tillage) all achieved better outcomes. The paper is well-written and solid.  The tables are clear. 

The Discussion has a misnumbering of citations that must be fixed. It is particularly notable because it occurs where the author compares the results to a companion study of fruit tree yields from the same work.  The citation is to paper 40, but clearly the author intends to reference paper 42.  The other references appear frameshifted as well.  Altogether though, this appears to be a solid contribution that addresses major weed suppression methods in an important agricultural context.

Minor comments:

Line 77-96: The hash marks make these sentences hard to read.  Please use numbering or some other way to increase readability.

Line 116: "1-2-20" should perhaps be "1-1-20"

Line 124: "were" not "was"

Table 2 missing end parenthesis "(S)"

Table 8 legend:  Clarify statistical comparison as "Means within column"

Line 255 not clear what "constated" means, please rephrase

Author Response

Skierniewice, 16.05.2023

Response to the Reviewer's 2 comments

The author wish to thank the Reviewer for your valuable comments that will improve the quality of the reviewed manuscript " Diversity of Summer Weed Communities in Response to Different Plum Orchard Floor Management in-row” (agronomy- 2391774). I have completed the manuscript  according to the Reviewers comments. The text has been corrected or supplemented to take into account the comments of the Reviewer.

In response to the first comment, I would like to inform you that during the processing of the text in the editor's office, an error in the numbering of references was made. In the original MS Word template as well as in the original PDF version sent to the editor's office, the number of references was 42. In items 7, 38, 42, links to publications were misread as references. The Spacebar or Enter key were unnecessarily used (probably my fault). The wrong numbering was the cause of the wrong citation.

 Ad line 77-96. Treatments are listed in points and additional comments are provided. Other notes: Corrected as directed by reviewer.

Yours Sincerely

Jerzy Lisek

Reviewer 3 Report

The manuscript is interesting and makes an important contribution to science in its field. After minor corrections, the manuscript is suitable for publication.

In the introduction, the author could use other synonymous terms instead of "crops" in some places, e.g. line 25: “These crops are of great /---/” maybe better is: “These cultivated plants are of great /---/”; Line 39/40: “/---/ horticulture (apple 39 trees and vegetable crops) is greater /---/” maybe better is: “/---/ horticulture (apple trees and vegetable) is greater /---/”; etc.

Line 33: You write: “/---/ providing food for bees and other pollinators /---/”. But I checked that (Prunus domestica) cultivar 'Valjevka' is self-pollinating. You should also mention that. If the 'Valjevka' variety does not need pollination by insects, was that the reason why you did not appreciate the promotion of the habitats of pollinating insects in the fruit tree garden?

The introduction should also include references to the EU's general agricultural policy. Today, the general EU agricultural policy is moving in the direction must be left more wild plants to grow in the agricultural landscape: https://agriculture.ec.europa.eu/common-agricultural-policy/income-support/greening_en

In the method chapter, should the author indicate which botanical Key book was used to determine the plants? In other words, where did the author get the names of the plant species. I checked that some Latin plant names differ from, for example, this international database of plant names:https://powo.science.kew.org/ . Was the Polish national botanical Key book used, if so which one? The method must also state who identified the plant species in these experiments?

In the method chapter, you must also state how large an area (Hectare? Two hectares?) and how many plum trees (10 pieces? 100 pieces?) were planted in the experimental field? What surrounds this experimental area: large cultivated fields, forests, natural open areas, water bodies? This affects the species richness, because if the natural species richness is low nearby (large cultivated fields all around), then plant seeds will not spread from there either.

It would add strength to the manuscript if there were photos of the experiment. For example, a photo from the first year, the last year, photos from different technologies, etc.

Line 275-277: “Valuable species may be more easily found if research is conducted at various locations across the region, close to semi-natural habitats such as South African vineyards [32].”

Reviewer viewed this reference 32: Steenwerth, K. L., Calderón-Orellana, A., Hanifin, R. C., Storm, C., & McElrone, A. J. (2016). Effects of various vineyard floor management techniques on weed community shifts and grapevine water relations. American Journal of Enology and Viticulture67(2), 153-162.

But I didn't find that fact stated in the article. Is there a wrong reference [32] in the manuscript behind this fact?

Author Response

Skierniewice, 16.05.2023

Response to the Reviewer's 3 comments

The author wish to thank the Reviewer for your valuable comments that will improve the quality of the reviewed manuscript " Diversity of Summer Weed Communities in Response to Different Plum Orchard Floor Management in-row” (agronomy- 2391774). I have completed the manuscript  according to the Reviewers comments. The text has been corrected or supplemented to take into account the comments of the Reviewer.

Ad comment „In the introduction, the author could use other synonymous terms instead of "crops" in some places, e.g. line 25: “These crops are of great /---/” maybe better is: “These cultivated plants are of great /---/”; Line 39/40: “/---/ horticulture (apple 39 trees and vegetable crops) is greater /---/” maybe better is: “/---/ horticulture (apple trees and vegetable) is greater /---/”; etc.

Done

Ad comment line 33: “You write: “/---/ providing food for bees and other pollinators /---/”. But I checked that (Prunus domestica) cultivar 'Valjevka' is self-pollinating. You should also mention that. If the 'Valjevka' variety does not need pollination by insects, was that the reason why you did not appreciate the promotion of the habitats of pollinating insects in the fruit tree garden?”

In English, the word "pollinator" means both the variety necessary to pollinate another and the insect that carries the pollen.  ‘Valijevka’ is a self-pollinating cultivar, i.e. that it does not need the pollen of other cultivar for pollination, unlike most varieties of apple, pear and cherry trees. However, better fruit setting is observed if pollen is transferred between flowers, which is done by insects. When it comes to the benefits of weeds to pollinators (insects), it's not just that insects are essential for pollinating trees. Weeds are a food source for pollinating insects when cultivated plants are not in bloom. And the flowering period of crops is relatively short and occurs mainly in spring and early summer. The mention of the benefits resulting from weeds to pollinating insects is therefore most advisable. 

Ad comment “The introduction should also include references to the EU's general agricultural policy. Today, the general EU agricultural policy is moving in the direction must be left more wild plants to grow in the agricultural landscape: https://agriculture.ec.europa.eu/common-agricultural-policy/income-support/greening_en”

Done

Ad comment “In the method chapter, should the author indicate which botanical Key book was used to determine the plants? In other words, where did the author get the names of the plant species. I checked that some Latin plant names differ from, for example, this international database of plant names:https://powo.science.kew.org/ . Was the Polish national botanical Key book used, if so which one? The method must also state who identified the plant species in these experiments?”

The Polish national botanical Key book was used. I added it to the references list. I also updated the Latin plant names, according to the cited key. The author identified the weeds.

Ad comment “In the method chapter, you must also state how large an area (Hectare? Two hectares?) and how many plum trees (10 pieces? 100 pieces?) were planted in the experimental field? What surrounds this experimental area: large cultivated fields, forests, natural open areas, water bodies? This affects the species richness, because if the natural species richness is low nearby (large cultivated fields all around), then plant seeds will not spread from there either.”

I added the necessary information. A homogeneous block of plum orchard had an area of ​​2 ha (about 2500 trees).

Ad comment “It would add strength to the manuscript if there were photos of the experiment. For example, a photo from the first year, the last year, photos from different technologies, etc.”

I agree that the photos are a valuable support for the article. Unfortunately, the photos that I have completed in the documentation of the experiment are not of good quality.

Ad “wrong references numbering

I would like to inform you that during the processing of the text in the editor's office, an error in the numbering of references was made. In the original MS Word template as well as in the original PDF version sent to the editor's office, the number of references was 42. In items 7, 38, 42, links to publications were misread as references. The Spacebar or Enter key were unnecessarily used (probably my fault). The wrong numbering was the cause of the wrong citation.

Yours Sincerely

Jerzy Lisek

Reviewer 4 Report

Row 9: after "years" add "from 2009 to 2015"; Row 72-73: provide value of the  average annual air temperature; row: 100: "data were collected between 20 July and 10 August", please indicate that it was during the investigation period, annualy, each year from 2008 to 2015; in Tables: 1, 3, 4, 5, 6: table title: "in the base year and in seventh year of OFM iplementation", consider to change: "in the base year (2008) and in the last year of OFM implementation (2015)"; Chapter References, please check that journal names are abbreviated, row 407 "Ecological Indicators" is "Ecol. Indic.", row 452: "Comunicata Scientiae" is "Comun. Sci".

The paper presents relevant facts, based on the multi-annual field work, combined with botanical research, and selection of various weed managament practice and measures in the plum orchard.  The authors applied relevant methodologiy and statistical approach. In the last year of the field work, the share of grasses in the weed cover was higher in all treatments. Shannon-Wiener Diversity Index and Simpson Dominance Index were calculated, compared and analysed. The main contribution is that reduction in the weed cover was achieved after the use of mulch, and such treatment is not harmful to the environment and avoids use of the chemicals in the orchards. Chapter: Discussion gives sufficient comparison with other relevant researches, with appropriately selected references. Chapter: Conclusion is well written.

The english language quality is acceptable.

Author Response

Skierniewice, 16.05.2023

Response to the Reviewer's 4 comments

The author wish to thank the Reviewer for your valuable comments that will improve the quality of the reviewed manuscript " Diversity of Summer Weed Communities in Response to Different Plum Orchard Floor Management in-row” (agronomy- 2391774). I have completed the manuscript  according to the Reviewers comments. The text has been corrected or supplemented to take into account the comments of the Reviewer.

Ad detailed comments Row 9: after "years" add "from 2009 to 2015"; Row 72-73: provide value of the  average annual air temperature; row: 100: "data were collected between 20 July and 10 August", please indicate that it was during the investigation period, annualy, each year from 2008 to 2015; in Tables: 1, 3, 4, 5, 6: table title: "in the base year and in seventh year of OFM iplementation", consider to change: "in the base year (2008) and in the last year of OFM implementation (2015)"; Chapter References, please check that journal names are abbreviated, row 407 "Ecological Indicators" is "Ecol. Indic.", row 452: "Comunicata Scientiae" is "Comun. Sci".

The text has been corrected or supplemented to take into account the comments of the Reviewer.

Yours Sincerely

Jerzy Lisek

Round 2

Reviewer 1 Report

The authors have done the suggestions to improve the manuscript.